# The Interaction of Fungicide and Nitrogen for Aboveground Biomass from Flag Leaf Emergence and Grain Yield Generation under Tan Spot Infection in Wheat

**DOI:** 10.3390/plants12010212

**Published:** 2023-01-03

**Authors:** Matías Schierenbeck, María Constanza Fleitas, María Rosa Simón

**Affiliations:** 1Genebank Department, Leibniz Institute of Plant Genetics and Crop Plant Research (IPK), OT Gatersleben, Corrensstr 3, 06466 Seeland, Germany; 2Cereals, Faculty of Agriculture and Forestry Sciences, National University of La Plata. Av.60 y 119, La Plata 1900, Argentina; 3CONICET CCT La Plata. Calle 8 Nº 1467, La Plata 1900, Argentina; 4Department of Plant Sciences, College of Agriculture and Bioresources, University of Saskatchewan, 51 Campus Drive, Saskatoon, SK S7N 5A8, Canada

**Keywords:** yield, crop growth rate, green leaf area, fungal diseases, *Pyrenophora tritici-repentis*, integrated management, chemical control, fertilizers

## Abstract

*Pyrenophora tritici-repentis* (Died.) Drechs., the causal agent of tan spot, is one of the most serious biotic diseases affecting wheat worldwide (*Triticum aestivum* L.). Studying the interaction between different fungicide mixtures and nitrogen (N) rates under tan spot outbreaks is of key importance for reducing aboveground biomass and grain yield losses. Taking this into account, our study took a mechanistic approach to estimating the combined effect of different fungicides and N fertilization schemes on the severity of tan spot, green leaf area index, SPAD index, aboveground biomass dynamics, and yield in a wheat crop affected at the reproductive stage. Our results indicated that reductions in green leaf area, healthy area duration (HAD), and the chlorophyll concentration (SPAD index) due to increases in the percentage of damage led to decreases in biomass production (−19.2%) and grain yield (−48.1%). Fungicides containing triazole + strobilurin + carboxamides (TSC) or triazole + strobilurin (TS) combined with high N doses showed the most efficient disease control. The positive physiological effects of TSC fungicides, such as extending the green leaf area, are probably responsible for the greater production of aboveground biomass (+29.3%), as well as the positive effects on grain yield (+15.8%) with respect to TS. Both fungicide treatments increased grains per spike, kernel weight, spikes m^−2^, grains m^−2^_,_ and grain yield. The increase in biomass in the TSC tended to cause slighter non-significant increases in grains per spike, 1000-kernel weight and grain yield compared with TS. The linear regression revealed positive associations among the extension of HAD and biomass (+5.88 g.m^−2^.HAD^−1^.day^−1^), grain yield (+38 kg.ha.HAD^−1^.day^−1^), and grain number (100.7 grains m^2^.HAD^−1^.day^−1^), explained by the interactions of high N doses and fungicides. Our study is the first report of the positive effect of TSC fungicides with high N doses on grain yield related-traits under tan spot infections in wheat.

## 1. Introduction

Future production of bread wheat is strategic to ensure worldwide food security [1]. It is considered the leading source of vegetable protein and provides around 20% of global calories for human consumption. Nitrogen (N) is key for cereal production, enabling increased growth and higher biomass, consequently influencing grain yields as well as the grain protein concentration, which is essential for quality and milling attributes [2]. Many biotic and abiotic stresses can affect wheat’s productivity. Abiotic stresses have received the most attention in recent literature. It has been demonstrated that artificial selection improves drought tolerance in hexaploid species more than in tetraploid species. However, when drought stress exceeded a certain threshold, both species suffered and applied a similar survival mechanism. Furthermore, under drought stress, hexaploid species had higher grain yield, grains per spike, grain weight, aboveground biomass, harvest index, and water use efficiency than tetraploid species [3,4]. Genes inducing thermotolerance in plants have also been discovered [5], and the role of jasmonic acid in the morphological, biochemical, and genetic levels of salt-stressed and non-salt-stressed wheat plants, as reflected in their yield attributes, has been clarified [6]. Furthermore, while the roots’ functional traits determine the efficiency of pre-anthesis N uptake, the efficiency of using the N taken up before anthesis to produce grains can be influenced by a variety of factors, including the presence of arbuscular mycorrhiza (AMF) and pathogens [7]. Roots’ association with arbuscular mycorrhiza reduced N remobilization efficiency in varieties with a high efficiency of using pre-anthesis N for grain production. Furthermore, as a result of a significant increase in post-anthesis N uptake aided by AMF and/or other microbes, the overall grain N concentration increased [7]. When exposed to heavy metals, AMF are also beneficial to plant growth [8].

Yield and flour quality are usually threatened by foliar diseases such as tan spot, which is caused by *Pyrenophora tritici-repentis* (Died.) Drechs and its anamorph, *Drechslera tritici-repentis* (Died.) Shoem., particularly in environments with high infection pressure caused by temperate conditions, adequate moisture, and rotations dominated by cereals [9,10,11]. Although the pathogen can affect the crop at any stage of growth, the key period for determining grain number was set close to the date of anthesis [12]. Grain number is considered to be the main driver explaining variations in grain yield, with grain weight having a minor impact [13,14]. The number of spikes.m^−2^ is usually affected by early epidemics of foliar diseases [15,16,17], which are frequently linked to necrotrophic pathogens when the inoculum is present in crop residues, as well as favorable conditions for disease growth in the early stages, especially in vulnerable cultivars. Furthermore, GPS and TKW are usually affected by most foliar diseases [18,19,20,21]. However, TKW is not reduced in some late infections for a variety of reasons, including trade-offs between TKW and low grain number, resulting in a higher sink/source ratio (accumulated absorbed radiation/grain number). Another factor that could explain this effect is the compensation caused by other organs (e.g, spikes and stems) under reductions in leaf area caused by fungal disease epidemics [22,23].

Although natural infections can reduce yield by up to 53% [24,25] due to decreases in the components of grain yield [26], the effect of artificial infections of tan spot and its interaction with diverse N schemes and pesticides mixtures on GLAI dynamics, biomass production, grain yield, and its components under field conditions has seldom been documented.

In places with predisposing conditions for epidemics and cultivars lacking acceptable levels of resistance to leaf spot, such as the Argentinian wheat belt, chemical control is regarded as an important strategy for disease management [27,28,29]. Flag leaves provide up to 58% of overall photosynthetic activity and 43% of the photosynthates required during grain-filling in bread wheat [30]. Fungicides can reduce yield losses by increasing the absorption of radiation, assimilate formation, and partitioning during grain-filling under disease pressure [31,32]. However, some *D. tritici-repentis* strains have been reported to be resistant to triazole and strobilurins. A mutation of G143A conferring QoI resistance to South American isolates of *Pyrenophora tritici-repentis* has been found [33]. Several strobilurins (azoxystrobin, trifloxystrobin, and pyraclostrobin) inhibited mycelial growth less effectively. Triazoles (epoxiconazole, propiconazole, tebuconazole, and prothioconazole) also reduced the isolates’ sensitivity, with prothioconazole being the most effective [34]. For that reason, new molecules should be incorporated as a management strategy for tan spot. According to several researchers, including succinate dehydrogenase inhibitors (SDHIs molecules or carboxamides) in mixtures of double demethylation inhibitors (DIMs or triazole) + quinone outside inhibitors (QoI or strobilurins) in wheat and barley has resulted in better control of fungal diseases [21,35]. Carboxamide treatments in crops have been shown to have positive physiological benefits, such as delayed senescence of the ears and leaves, an increased photosynthetic rate, and increased water and radiation use efficiency [31,36,37].

During the last 60 years, the use of N fertilization has increased due to a combination of new agronomic techniques and the deployment of semi-dwarf genotypes with higher N use efficiency [2]. Furthermore, N fertilizer use might influence pathogen–plant interactions. In this sense, the plants’ N nutrition status has differing responses to disease severity, depending on numerous factors such as the climatic conditions, the genotype, and the nature of the agent affecting the wheat crops [38]. Previous reports suggested that low N availability could affect plants’ susceptibility to necrotrophic fungi such as *P. tritici-repentis*. This response is due to a decrease in plant vigour and increases in the pathogen’s activity, as it colonizes weak tissues more effectively [39]. The rate, timing, and form of N applied have also been reported to affect the severity of tan spot [21,40].

Our study took a mechanistic approach to estimating the combined effect of fungicides and N doses on the severity of tan spot, GLAI, the SPAD index, aboveground biomass dynamics, and yield [41]. We hypothesized that the addition of carboxamides to a triazole–strobilurin mixture could interact with N schemes. Thus, we expected the most efficient disease control would occur for combined fungicide mixtures, including carboxamides and high N doses. This effect would lead to an improvement in ecophysiological traits, yield, and yield components.

## 2. Materials and Methods

### 2.1. Location and Environmental Conditions

Field trials were performed during two crops cycles (2014 and 2015) at the Faculty of Agriculture and Forestry Sciences, National University of La Plata, Argentina. Analysis of the soil samples indicated the following values: organic matter, 3.60%; N, 0.201%; phosphorus, 28.4 mg.kg^−1^; pH, 5.81. Meteorological data during the crop cycle were collected and contrasted with the historical data (Figure 1a,b). 

### 2.2. Treatments

The fungicide treatments were the main plots: (1) untreated (without fungicide), (2) TS (BASF Opera^TM^: epoxiconazole (50 g.L^−1^) + pyraclostrobin (133 g.L^−1)^), and (3) TSC (BASF Orquesta Ultra^TM^: fluxapyroxad (50 g.L^−1^) + epoxiconazole (50 g.L^−1^) + pyraclostrobin (81 g.L^−1^)). Three doses of N fertilization applied as urea (kg.ha^−1^), namely 0 kg N.ha^−1^, 70 kg N.ha^−1^, and 140 kg N.ha^−1^, divided between sowing time and the stem elongation growth stage (GS 31) [42], were the subplots. The widely used cultivar Baguette Premium 11, with moderate susceptibility to tan spot and a mid–late heading date was selected for the experiments. Each sub-subplot was 5.5 m by 1.4 m (7.7 m^2^).

### 2.3. Inoculum Preparation and Fungicide Treatments

Isolates of the pathogen were grown on V8^®^ media using the protocol suggested by [43]. Using a Neubauer haemocytometer, the inoculum suspension was adjusted to 3000 spores.mL^−1^. The plots were sprayed until runoff at the two-tiller stage (GS 22) and GS 31 [11] to generate an infection at the reproductive stage. To reduce the interference of *Puccinia triticina*, which causes leaf rust, a specific Basidiomycota fungicide (CHEMTURA Plantvax^TM^, 75% oxycarboxin, dose = 750 g.ha^−1^) was applied at GS 39 (the flag leaf stage) and the beginning of anthesis (GS 61). Double and triple fungicide mixtures were sprayed at the three-tiller stage (GS 23) and GS 39 at doses of 1.0 L.ha^−1^ (TS) and 1.2 L.ha^−1^ (TSC). Infections started during shooting (GS 32–GS 33) and were visualized after emergence of the flag leaf; for that reason, severity assessments were carried out from GS 39 onwards.

### 2.4. Measurements

Tan spot severity was determined on seven randomly selected plants per plot by visually estimating the percentage of leaf area damaged at GS 39, GS 61, and GS 82 (dough stage) on all leaves with at least a portion of green tissue.

At the same stages, measurements were made on seven flag leaves with a chlorophyll meter (Minolta SPAD-502) to obtain an indirect indicator of the level of chlorophyll and an estimator of the chlorosis and/or necrosis generated by the presence of pathogens [44].

The total leaf area index was assessed in the same three stages by counting the tillers within 0.5 m^2^ and quantifying the size of the leaves using rulers. From these calculations and the tan spot severity (%), the green leaf area index (GLAI) was calculated, and the healthy area duration (HAD) was determined by following the formula proposed by [45].

For calculating the aboveground biomass (AGB; g.m^−2^), a 2 m row was sampled at GS 39, GS 60, and ripening (GS 95). For the determination of grain yield, a 1.1 m^2^ sample was harvested and threshed. Using this sample, the spikes per m^2^ (SPKN), grains per spike (GPS), grains per m^2^ (GN), and 1000-kernel weight (TKW) were determined.

### 2.5. Statistical Analysis and Experimental Design

A split-split-plot was the experimental design was utilized. An ANOVA was performed using GenStat software [46,47,48]. The factors were the years (2014 and 2015), fungicides (untreated, TS, and TSC), the N treatments (0, 70 and 140 N), and the three replications (as blocks). Means were compared via the LSD test (*p* ≤ 0.05). 

## 3. Results

### 3.1. Meteorological Data

Precipitation from June to December (the crop cycle) was higher in both years than the historical average (523 mm), but it was much higher in 2014 than in 2015. The mean temperatures were also slightly higher in 2014 (Figure 1a,b).

### 3.2. Disease Severity, Green Leaf Area Dynamics, and Aboveground Biomass Production

Meteorological conditions influenced the development of tan spot disease, resulting in significantly higher values in the first year of assessment at all three stages evaluated, which were associated with the high precipitation values. Even though it was a field experiment exposed to several pathogens, the majority of the damage was caused by tan spot. This was due to the pathogen’s high inoculum pressure and the application of a specific fungicide for controlling leaf rust, the other important disease. Furthermore, because the experiments were far from wheat production areas, the natural inoculum was low. 

On average and compared with the untreated control, TS/TSC fungicides reduced the severity (percentage of damage) by up to 3.9% (GS 39), 28.2% (GS 60), and 16% (GS 82). The highest N doses decreased severity by 1.7% (GS 39), 10.1% (GS 60), and 10.8% (GS 82) compared with the 0 N treatment (Table 1, Table 2 and Table 3). The fungicide × N interaction revealed that the highest N rates among all the fungicides evaluated resulted in significant reductions in tan spot severity at the three stages. In particular, for the severity at GS 39 and GS 60, greater decreases caused by the N doses were observed in the untreated control. When the untreated–0 N and TSC–140 N treatments were contrasted at the different growth stages, the severity decreased by 5.9% (GS 39), 38.9% (GS 60), and 24.5% (GS 82) (Figure 2a).

The GLAI dynamics and the HAD were affected by the meteorological conditions, showing higher values during 2014 due to the higher precipitation in comparison with 2015. Fungicide treatments and increased N fertilization resulted in significant increases in GLAI at all the growth stages studied and also increases in HAD (Table 2 and Table 3). Higher N doses caused increases in HAD of up to 90% (untreated), 61% (TS), and 70% (TSC) when the 0 N and 140 N doses were compared (data not shown). Fungicides containing TS and TSC combined with the 70 N and 140 N dose rates increased GLAI at the three growth stages as well as HAD, with the highest values for the 140 N–TSC treatments. In this sense, comparisons between extreme treatments (untreated–0 N vs. TSC–140 N) at the different growth stages showed differences in GLAI (m^2^ leaf per m^2^ soil) of +5.73 (GS 39), +4.42 (GS 60), and +1.05 (GS 82) (Figure 2b).

SPAD values at the three growth stages showed differences caused by the fungicides, N doses, and fungicide × N interaction (Table 3). Fungicides increased SPAD by 26% and 29.4% for TS and TSC, respectively, at GS 39; by 35.5% and 40.3% at GS 60; and by 58.5% and 70.3% for TS and TSC, respectively, at GS 82 relative to the untreated control, while no variations were detected among the fungicide mixtures. Moreover, higher N doses increased the SPAD values by up to 17.2% during GS 39, 12.8% at GS 60, and 37.4% at GS 82. The fungicide × N interaction presented higher values when TS and TSC fungicides were combined with increases in the N dose. For all growth stages, greater increases in the SPAD values were observed with the high N dose in untreated treatments compared with 0 N. Moreover, increases of up to 49.9% (GS 39), 58.2% (GS 60), and 163.7% (GS 82) were detected when the untreated–0 N and TSC–140 N treatments were compared at the different growth stages (Figure 2c).

For aboveground biomass production, the highest N doses at all the stages evaluated produced increases of up to 28.2% (GS 39), 32.3% (GS 60), and 29.3% (GS 95) for 140 N compared with 0 N, and up to 21.3% (GS 39), 17.6% (GS 60), and 15.9% (GS 95) when the 70 N and 0 N treatments were contrasted (Figure 3a; Table 4 and Table 5). The application of TS and TSC fungicides improved biomass production by 11.6% and 16.2% (GS 39), 9.6% and 14.6% (GS 60), by 12.2% and 19.2% (GS 95), respectively, compared with the untreated plots (Figure 3b; Table 5).

### 3.3. Yield and Main Components

Differences between the years were detected for some of the yield components evaluated. In this sense, in 2014, there were increases in GPS (+8%), SPKN (+15.6%), and GN (+25.3%) compared with 2015 (Table 4 and Table 5).

For their part, the TS and TSC fungicide mixtures improved the grain yield components, but no differences were detected between these mixtures. Compared with the untreated control, TKW increased by up to 21.5%, GPS increased by +14.8%, SPKN increased by +11.7%, and GN increased by +29.2% when fungicide mixtures were applied (Table 5).

Nitrogen doses and the year × N interaction affected grain yield and the yield components. In general terms, the 70 N and 140 N doses showed increases of up to 6.4% in GPS, 14.1–23.7% in SPKN, 16.7–31.5% in GN, and 14.5–24.1% in GY compared with 0 N, without significant differences between TS and TSC. When the year × N interaction was analysed, GPS revealed non-significant responses in 2014 and increases of up to 11.6% in 2015. For GN, increases of up to 38.5% and 23.4% were reported for 2014 and 2015, respectively. The TKW did not show significant responses in 2014, while decreases of 5.2% were reported under high N doses. For their part, SPKN increased by up to 36.4% (2014) and 10.8% (2015) (0 N vs. 140 N).

The fungicide treatments produced increases in grain yield of up to 42.3% (TS) and 48.2% (TSC) in comparison with the control, whereas N doses improved it from 14.5% (70 N) to 24.1% (140 N) compared with the non-fertilized plots (Table 1 and Table 3). Increases in N doses in the untreated plots did not show significant effects when the fungicide × N interaction was analysed. For their part, the combination TS–140 N enhanced yield by 15.7%, TSC–70 N increased it by 22.8%, and TSC–140 N improved this trait by up to 42.7% compared with 0 N, with the increase under TSC–140 N being significantly higher than that of the other treatments (Figure 4). Higher rates of increase in grain yield were reported for TSC fungicides across different N schemes (39.02 kg.ha^−1^.day^−1^) compared with TS fungicides (20.4 kg.ha^−1^.day^−1^) and without fungicides (11.4 kg.ha^−1^.day^−1^) when a regression analysis between grain yield and HAD was performed (data not shown).

### 3.4. HAD Acts as an Indicator of the Positive Effects Generated by Fungicide Mixtures and N Fertilization under Tan Spot Epidemics

The effect of disease management tools such as N fertilization and fungicides on HAD and different physiological traits related to AGB and yield generation were assessed using regression analysis for the fungicide × nitrogen interaction (Figure 5). The combinations of high N doses (140 N) + TSC fungicide produced important extensions of HAD that led to increases in AGB, SPAD, grain yield, and its components, reversing the negative effects caused by tan spot epidemics. In this sense, an important negative association was detected between increases in disease severity at GS 82 and HAD, showing a rate of −3.82 HAD day for a 1% increase in severity, while a positive response was detected for SPAD at GS 82 and extensions of HAD caused by the highest N doses and their interaction with TS/TSC fungicides (+0.208 SPAD value.HAD^−1^.day^−1^) (Figure 5a,b).

The accumulation of aboveground biomass at harvest (+5.88 g.m^−2^.HAD^−1^.day^−1^), as well as grain yield (+38.01 kg.ha.HAD^−1^.day^−1^), were positively correlated with extensions of HAD (Figure 5c,d). Grain yield components such as spike number (1.74 spikes m^2^.HAD^−1^.day^−1^), grains per spike (0.067 grains per spike.HAD^−1^.day^−1^), and grain number (100.7 grains m^2^.HAD^−1^.day^−1^) were also highly and positively correlated with extensions of HAD as a result of management practices, while TKW did not show any associations with HAD (Figure 5e–h).

## 4. Discussion

In this work, the effects of a combination of three fungicides with different N doses under high *P. tritici-repentis* disease pressure on green leaf area dynamics, aboveground biomass generation at the reproductive stage, yield, and yield components were assessed under field conditions in Argentina. 

The meteorological conditions were more conducive for tan spot development in 2014 than in 2015, explained by the higher temperatures, relative humidity, and rainfall accumulated during the crop cycle [25]. The results obtained in this work allowed us to establish that the N availability in wheat plants had a close relationship with the severity of tan spot, in line with [39]. Concerning the development of the disease, a marked decrease was found in the severity levels caused by increases in the N dose (70 N and 140 N) at all growth stages evaluated. These effects were in line with the data collected by [17,49] who reported the delayed development of tan spot under higher N doses. In this sense, the work carried out by [40] agrees regarding the suppressive effect that N has on this disease, based on increases in GLAI/HAD resulting from increases in N availability. 

It has been reported that the delay in senescence contributes indirectly to a lower dispersion of the secondary inoculum produced by conidia in maturing lesions on the leaves, as a result of increases in GLAI [50]. In contrast and coinciding with our results, low N availability has been associated with the greater potential inoculum of *P. tritici-repentis*, a greater epidemic rate, and anticipated crop senescence [40,51]. Moreover, it has been reported that decreases in N fertilization are closely related to higher susceptibility to necrotrophic pathogens, explained by a reduction in the defence capacity of plants [38]. Our results showed that high N fertilization rates (70 and 140 N) extended HAD, explained by decreases in the progress of tan spot disease and increases in the crop growth rate [52]. 

Fungicides are a fundamental tool in crop production for disease management in environments with high yield potential that are prone to fungal diseases. It has been suggested that the effective life of fungicides with diverse modes of action is likely to be maximized by applying them in mixtures [53]. This mechanism is explained by the reduction in the selection pressure on the pathogen, avoiding the develop of resistance to the active site of each single compound [35]. Fungicide mixtures have also been related to better disease control compared with protectants with a single active ingredient against Septoria leaf blotch [54] and leaf rust [29,55,56,57,58]. These findings are in line with the results reported here, where the TSC fungicide showed the lowest disease severity values, causing higher GLAI values across the whole crop cycle and consequently extending HAD with respect to the TS fungicides. Some authors have reported that carboxamides triggered a longer fungicide residuality that decreased fungal reinfection and induced a positive response in the plants’ physiology when it was included in fungicides with triazole + strobilurin [37,59]. Other authors have also verified the lower severity of tan spot and leaf rust by applying a triple mixture compared with a double TS combination for up to five weeks after application [35]. Moreover, extensions in crop HAD, particularly flag leaf HAD, were found when TSC combinations were sprayed in plots artificially inoculated with tan spot and leaf rust, as reported by [11]. 

Significant reductions in the SPAD leaf greenness index, an indicator of the chlorophyll content of the tissues, were detected under higher inoculum pressure. These results are in line with [60,61,62], who reported decreases in SPAD because of infections generated by several leaf pathogens. Similarly, [61] documented a negative correlation between the damage produced by *Bipolaris sorokiniana* and *P. tritici-repentis*, and the SPAD values. The negative effects of tan spot disease on the SPAD index were reversed following the fungicide spray and fertilization with 70 N/140 N. As the severity values decreased more under higher N doses in the untreated plots, the SPAD values showed higher increases in the untreated treatments with the high N dose. Moreover, the strong effect of fungicides containing triazole + strobilurin + carboxamide in combination with high N doses on SPAD values under late *P. tritici-repentis* infections (flag leaf stage onwards) in wheat were reported for the first time in this study.

Although the positive influence of fungicides on biomass production has been reported for TS mixtures [27,57,63], few reports are available showing the effects of TSC mixtures on AGB production and their interaction with different N schemes as reported in this work. The aboveground biomass production was increased at the three growth stages evaluated under applications of TS and TSC fungicidal molecules. Additionally, TSC mixture caused the greatest increase, as documented by [37]. Increases in biomass generation of up to 16.2% at GS 39, 14.6% at GS 60, and 19.2% in GS 95 were found the TSC fungicide was used compared with non-protected plots. In agreement with our results, increases in biomass production from fungicides use have been documented so far as a result of the reduction in the level of disease severity, delays in flag leaf senescence, and an enhanced crop growth rate. This response is explained by increases in the radiation use efficiency and absorption radiation [11,34,64]. A close relationship between biomass production and the amount of radiation absorbed by the canopy has been previously documented [12], so that the increases in GLAI and HAD detected under fungicide use would explain the higher AGB accumulation. Recently, several wheat genotypes were compared in agricultural fields under two different management practices: N rates sufficient for reaching the yield goals + no fungicide application, and high N rates + fungicides. The prevalent diseases early in the season were tan spot, Septoria, and powdery mildew, whereas leaf and stripe rusts were prevalent late in the season. The results indicated that AGB production was the primary driver of the grain yield responses, while changes in the grain number were the main component affected by the management practices evaluated [65].

In this work, biomass production was also dependent on the dose of N applied. Moreover, rising N rates caused increased AGB generation at GS 39, GS 60, and GS 95, showing significant differences among the 0 N, 70 N, and 140 N doses. In that sense, aerial biomass increased by up to 28.2% (GS 39), 32.3% (GS 60), and 29.3% (GS 95) when the 0 N and 140 N treatments were contrasted. In line with our results, [66] documented increases in aerial biomass resulting from increases in N fertilization in several European elite genotypes. Furthermore, the positive effect of N fertilization on biomass production is in close agreement with their positive effect on green leaf area and its duration (HAD), as well as the N accumulated in the leaves (SPAD index), as was documented in this work. In addition, it has been reported that GLAI and the N concentration in the leaves play a key role in enhancing the crop growth rate through their influence on the interception of radiation and radiation use efficiency [52].

The positive effects of the N dose × fungicide application interaction on lowering disease severity and enhancing parameters such as GLAI, HAD, AGB, and SPAD described here were the primary drivers for improvements in grain yield. The significance of this study rests on the findings on the effect of new protectant TSC mixtures compared with TS and its interaction with different N doses under artificial inoculations of *P. tritici-repentis* on the ecophysiological parameters related to AGB (GLAI, HAD, SPAD) as well as yield.

Regarding the impact of fungicides on grain yield, [28] reported that increases in wheat yields caused by the use of fungicides under tan spot epidemics depended on the amount of stubble infected, the severity of the disease, genotypic resistance, the fungicide use strategy, the type of active substance, the time and number of applications, and the doses and methods of application. In this crop, the period between spike growth and 10 days after anthesis is crucial for the determination of yield, since the grain number is determined at this point [12,13,14]. In this critical period, there is a strong source limitation, so GLAI/HAD extensions during this period are essential for providing assimilates to the spikes and increasing their fertility [58]. Taking this into account and in agreement with our results, the incidence of foliar diseases during this period will trigger yield losses caused by decreases in the GN or/and TKW [67].

Moreover, positive correlations between HAD and grain yield were reported when the fungicide treatments were evaluated with increases in the N dose. Compared with the untreated control, fungicides containing triazole + strobilurin + carboxamide showed the highest grain yield increases, followed by the TS treatment; similar responses were previously reported by [37,59] in the absence of disease pressure. Moreover, in disease-free experiments, it has been reported that carboxamides enhance several physiological traits of plants such as a retardation of spike and leaf senescence after anthesis, and increases in water and radiation use efficiency as well as in the photosynthetic rate [31,59]. Furthermore, other authors have documented that these fungicides produced yield increases as a result of increases in the PSII photosystem efficiency and extensions of HAD [37,68]; these responses coincided with the higher SPAD index values reported in our work under TSC mixtures. For their part, [69] reported increases in wheat yields of up to 2 t.ha^−1^ when combinations of higher N doses + TS molecules were evaluated under infections of pathogens such as Fusarium head blight, leaf rust, Septoria, and powdery mildew. Recently, over a range of cultivars infected with *P triticina*, under different fungicides and N rates in Argentina, TSC mixtures produced greater control of leaf rust (−15.3% of AUDPC), which translated into a greater total leaf area index (+13.3%), extended HAD (+8.3%), and ultimately had the highest grain yields (+12.1%) when compared with the TS protectant [56]. In this sense, some reports [69,70,71] have indicated that the use of TS mixtures delayed green leaf area senescence, leading to higher radiation absorption and a longer grain-filling period. The superior effect of fungicides that included carboxamides (TSC) in comparison with TS and their interaction with various N fertilization schemes on ecophysiological traits (GLAI, HAD, SPAD index, disease severity) related to aboveground biomass production and grain yield under *P. tritici-repentis* infections is the main novelty of our study.

## 5. Conclusions

This study showed that the interaction between different N schemes and chemical controls are appropriate for reversing the harmful effects of *P. tritici-repentis* epidemics on traits related to aboveground biomass and grain yield generation. Negative effects caused by epidemics of this pathogen starting at shooting can be explained by increases in the disease severity and decreases in GLAI and the SPAD dynamics, leading to reductions in AGB and grain yield. Chemical control in combination with high N rates can reverse these effects, which was detected through the positive correlations between extensions of HAD and several variables related to biomass generation and yield. The efficient disease control plus the improvement in several physiological traits produced by TSC fungicides in combination with the 140 N dose could explain the higher gains in GLAI, HAD, AGB, and grain yield with respect to TS fungicides at the same N levels. A better response in the variables related to grain yield generation under tan spot epidemics was reported when TSC fungicides were combined with high N rates.

## Figures and Tables

**Figure 1 plants-12-00212-f001:**
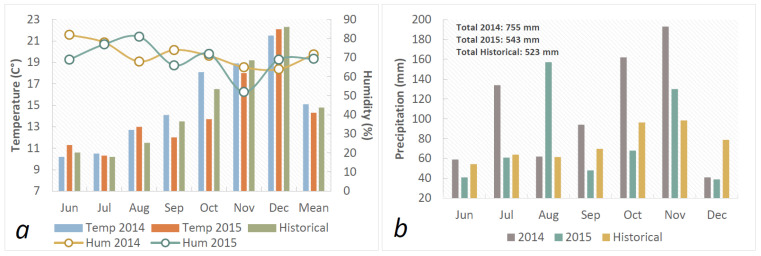
Environmental conditions during 2014 and 2015. (**a**) Mean temperature (°C) and humidity (%); (**b**) Precipitation.

**Figure 2 plants-12-00212-f002:**
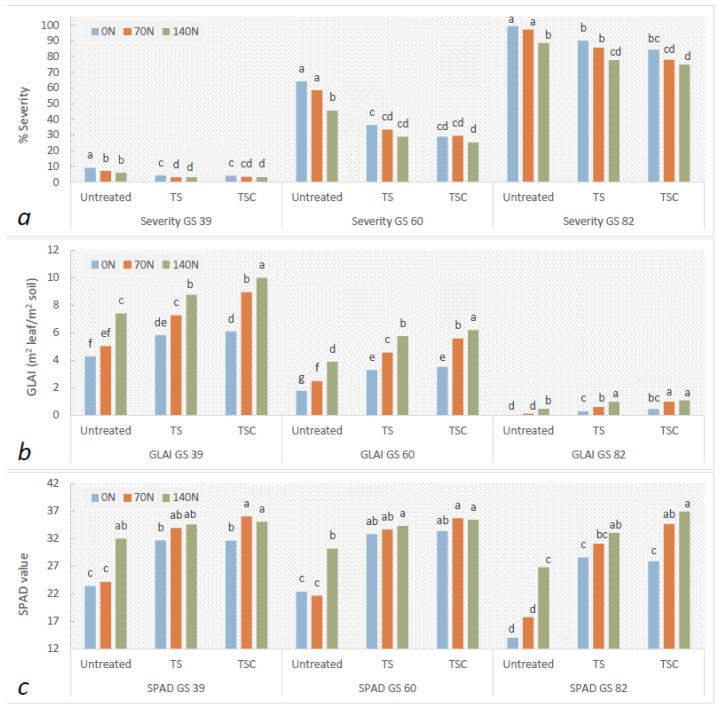
Means of (**a**) severity, (**b**) green leaf area index (GLAI), and (**c**) SPAD values for the fungicide × N treatment interaction at three growth stages (GS39, GS60, and GS82). Matching letters at the same growth stage are not statistically different (LSD *p* ≤ 0.05).

**Figure 3 plants-12-00212-f003:**
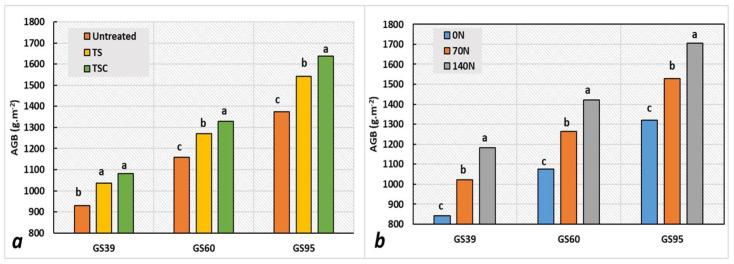
Means of aboveground biomass (AGB) at three growth stages (GS 39, GS 60, and GS 95) for (**a**) the fungicide treatments and (**b**) N dose treatments. Matching letters within same growth stages are not statistically different (LSD *p* ≤ 0.05).

**Figure 4 plants-12-00212-f004:**
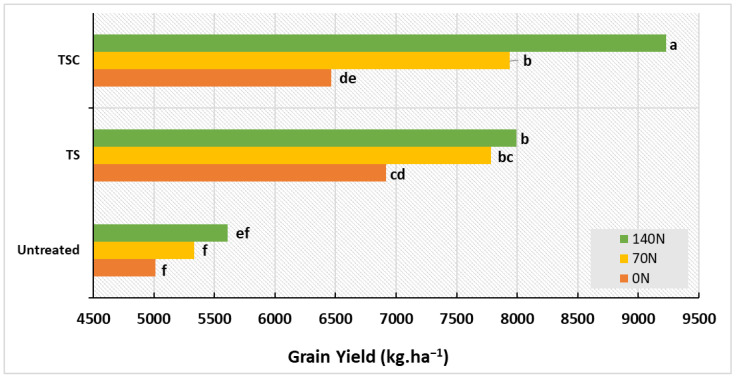
Means of grain yield for the fungicide × N interaction. Matching letters are not statistically different (*p* ≤ 0.05).

**Figure 5 plants-12-00212-f005:**
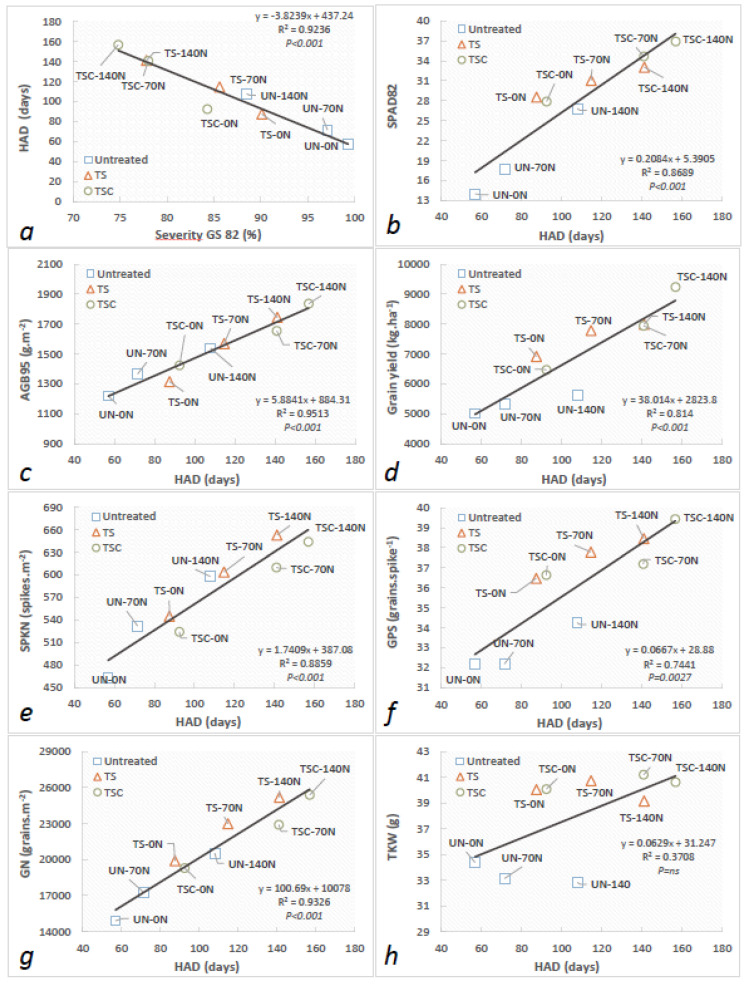
Linear regression of HAD (healthy area duration) with (**a**) severity at GS 82, (**b**) SPAD at GS 82, (**c**) aboveground biomass at GS 95 (AGB95), (**d**) grain yield, (**e**) spikes per m^2^ (SPKN), (**f**) grains per spike^1^ (GPS), (**g**) grains per m^2^ (GN), and (**h**) 1000-kernel weight (TKW). Points represent the means of the interaction between the N dose and fungicide for three replications.

**Table 1 plants-12-00212-t001:** ANOVA of severity (%SEV), green leaf area (GLAI), and healthy area duration (HAD) of bread wheat under different combinations of fungicides and N doses.

Source of variation	D.F.	%SEV39	% SEV60	%SEV82	GLAI39	GLAI60	GLAI82	HAD
**Years (Yr)**	1	111 (0.009)	2861 (<0.001)	858 (0.005)	10.8 (0.002)	1.01 (<0.001)	0.11 (0.003)	261.(0.004)
**Error A**	2	0.97	0.19	4.41	0.02	0.0003	0.0003	1.00
**Fungicide (Fun)**	2	88.4 (<0.001)	4061 (<0.001)	1181 (0.001)	35.3 (<0.001)	27.9 (<0.001)	1.88 (<0.001)	12,455 (<0.001)
**Yr × Fun**	2	7.92(0.004)	142 (0.339)	16.0 (0.796)	0.14 (0.892)	0.03 (0.687)	0.01 (0.748)	1.41 (0.988)
**Error B**	8	0.69	114	68.2	1.22	0.078	0.039	118
**Nitrogen (Ni)**	2	14.5 (<0.001)	472(<0.001)	536(<0.001)	49.9 (<0.001)	26.4 (<0.001)	1.52 (<0.001)	14,330 (<0.001)
**Yr × Ni**	2	1.30 (0.072)	16.5 (0.438)	9.86 (0.343)	0.19 (0.573)	0.03 (0.642)	0.009 (0.227)	3.20 (0.94)
**Fun × Ni**	4	2.48 (0.002)	93.8 (0.005)	14.9 (0.018)	1.86 (0.003)	0.72 (<0.001)	0.087 (<0.001)	445 (<0.001)
**Y × Fun *×* Ni**	4	0.22 (0.734)	3.28 (0.952)	10.6 (0.334)	0.007 (0.999)	0.0008 (1.000)	0.0005 (0.985)	0.20 (0.522)
**Error C**	24	0.44	19.3	8.81	0.35	0.67	0.006	51.9
**Total**	53							

**Table 2 plants-12-00212-t002:** ANOVA of the SPAD and aboveground biomass (AGB) dynamics of bread wheat under different combinations of fungicides and nitrogen doses.

Source of Variation	D.F.	SPAD39	SPAD60	SPAD82	AGB39	AGB60	AGB95
**Years (Yr)**	1	51.8 (0.462)	302 (0.206)	414 (0.102)	876,369 (<0.001)	1,877,710 (0.014)	2,834,048 (0.009)
**Error A**	2	63.6	88.8	49.4	8157	27,799	26,219
**Fungicide (Fun)**	2	322 (0.002)	543 (<0.001)	961 (<0.001)	108,715 (<0.001)	134,419 (<0.001)	319,933 (<0.001)
**Yr × Fun**	2	6.36 (0.742)	8.48 (0.602)	7.31(0.729)	8157 (0.191)	4007 (0.110)	19,475 (0.014)
**Error B**	8	20.5	15.6	22.2	3981	1358	2575
**Nitrogen (Ni)**	2	111 (<0.001)	71.3 (<0.001)	346 (<0.001)	514,680 (<0.001)	543,652 (<0.001)	673,632 (<0.001)
**Yr × Ni**	2	7.22 (0.353)	15.5 (0.081)	2.37 (0.657)	7882 (0.289)	3497 (0.466)	6718 (0.206)
**Fun × Ni**	4	35.1 (0.003)	37.9 (<0.001)	38.1 (<0.001)	1637 (0.893)	2122 (0.752)	6799 (0.181)
**Yr × Fun × Ni**	4	8.05 (0.331)	17.4 (0.033)	0.34 (0.993)	2024 (0.851)	686 (0.959)	7470 (0.147)
**Error C**	24	6.63	5.53	5.55	6031	4442	3978
**Total**	53						

**Table 3 plants-12-00212-t003:** Mean tan spot severity, SPAD, green leaf area at three growth stages (GLAI), and healthy area duration (HAD) of bread wheat under different combinations of fungicides and N doses.

	%SEV39	%SEV60	%SEV82	SPAD39	SPAD60	SPAD82	GLAI39	GLAI60	GLAI82	HAD
**Years**										
2014	6.21 a	46.2 a	90.1 a	30.4	28.7	25.1	7.52 a	3.98 b	0.51 b	110 a
2015	3.35 b	31.6 b	82.2 b	32.4	33.4	30.6	6.62 b	4.26 a	0.60 a	106 b
LSD	1.15	0.51	2.46	9.34	11.0	8.23	0.15	0.02	0.02	1.17
**Fungicide**										
Untreated	7.34 a	56.0 a	95.0 a	26.5 b	24.8 b	19.5 b	5.57 c	2.72 c	0.21 c	78.8 c
TS	3.47 b	32.9 b	84.5 b	33.4 a	33.6 a	30.9 a	7.28 b	4.54 b	0.62 b	114.2 b
TSC	3.53 b	27.9 b	79.0 b	34.3 a	34.8 a	33.2 a	8.35 a	5.10 a	0.84 a	130.1 a
LSD	0.64	8.22	6.35	3.48	3.04	3.62	0.85	0.22	0.15	8.39
**Nitrogen**										
0 N	5.79 a	43.1 a	91.2 a	28.9 c	29.5 b	23.5 c	5.39 c	2.86 c	0.26 c	78.9 c
70 N	4.47 b	40.5 a	86.9 b	31.4 b	30.4 b	27.8 b	7.09 b	4.22 b	0.57 b	109 b
140 N	4.08 b	33.2 b	80.4 c	33.9 a	33.3 a	32.3 a	8.72 a	5.28 a	0.84 a	135 a
LSD	0.46	3.03	2.04	1.77	1.62	1.62	0.41	0.18	0.05	4.95

TS, triazole + strobilurin mixture; TSC, triazole + strobilurin + carboxamide mixture. N doses (kg.ha^−1^): 0 N, 70 N, and140 N. Matching letters within the same variables are not statistically different (LSD *p* ≤ 0.05).

**Table 4 plants-12-00212-t004:** ANOVA of grains per spike (GPS), 1000-kernel weight (TKW), spikes per m^2^ (SPKN), grains per m^2^ (GN), and the yield of wheat under different fungicides and N rates.

Source of variation	D.F.	GPS	TKW	SPKN	GN	Grain Yield
**Years (Y)**	1	122 (0.010)	0.019 (0.987)	92,955 (<0.001)	2.106^7^ (<0.001)	114,030 (0.875)
**Error A**	2	1.19	56.2	44.1	1.879^8^	3,600,132
**Fungicide (Fu)**	2	138 (0.065)	285 (<0.001)	26,459 (0.014)	1.536^8^ (<0.001)	35,112,519 (<0.001)
**Y × Fu**	2	0.13 (0.996)	11.3 (0.433)	989 (0.756)	7.595^5^ (0.973)	380,523 (0.748)
**Error B**	8	35.2	12.1	3422	0.25	1,262,760
**Nitrogen (N)**	2	25.1 (0.004)	3.36 (0.352)	66,771 (<0.001)	1.454^8^ (<0.001)	9,987,390 (<0.001)
**Y × N**	2	11.3 (0.064)	11.3 (0.041)	20,797 (<0.001)	1.697^7^ (0.002)	854,157 (0.103)
**Fu × N**	4	1.36 (0.827)	3.20 (0.407)	644 (0.510)	8.897^7^ (0.773)	1,998,794 (0.002)
**Y × Fu × N**	4	1.36 (0.817)	3.20 (0.406)	646 (0.505)	1.110^6^ (0.695)	171,130 (0.734)
**Error C**	24	3.66	3.07	762	1.989^6^	340,526
**Total**	53					

**Table 5 plants-12-00212-t005:** Mean values of aboveground biomass (AGB; g.m^−2^) at different stages, 1000-kernel weight (TKW; g), grains per spike (GPS), spikes per m^2^ (SPKN), grains per m^2^ (GN), and grain yield (kg.ha^−1^) in wheat for different fungicides and nitrogen rates.

Source of Variation	AGB39	AGB60	AGB95	TKW	GPS	SPKN	GN	Grain Yield
**Year**								
2014	888 a	1067 b	1289 b	38.0	37.6 a	616 a	23,282 a	6966 a
2015	1143 a	1440 a	1747 a	38.0	34.6 b	533 b	18,573 b	6874 a
LSD	264	195	190	8.78	1.27	7.80	161	2221
**Fungicide**								
Untreated	929 a	1159 c	1375 c	33.4 b	32.9 b	531 b	17,555 b	5317 b
TS	1037 b	1271 b	1542 b	40.0 a	37.6 a	601 a	22,693 a	7563 a
TSC	1080 b	1329 a	1638 a	40.6 a	37.8 a	593 a	22,533 a	7878 a
LSD	48.50	28.3	39	2.68	4.56	45.0	4031	864
**Nitrogen**								
0N	843 c	1075 c	1319 c	38.2	35.1 b	510 c	18,029 c	6131 c
70N	1023 b	1263 b	1530 b	38.4	35.7 b	582 b	21,042 b	7018 b
140N	1081 a	1422 a	1706 a	37.5	37.4 a	631 a	23,710 a	7611 a
LSD	53.4	45.9	43.4	1.21	1.32	19.0	0.41	401

Triazole + Strobilurin mixture (TS); Triazole + Strobilurin + Carboxamide mixture (TSC); N doses (kg.ha^−1^): 0 N, 70 N, and 140 N. Matching letters within the same variables are not statistically different (LSD *p* ≤ 0.05).

## Data Availability

The data that support the findings of this study are available from the corresponding author upon reasonable request.

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
