# Peer review of "The Interaction of Fungicide and Nitrogen for Aboveground Biomass from Flag Leaf Emergence and Grain Yield Generation under Tan Spot Infection in Wheat"

_plants, 2023, doi:10.3390/plants12010212_

Round 1
Reviewer 1 Report
The manuscript presents some exciting novel results that should be published after some editing.This paper addresses research on the combined effects of fungicides and nitrogen fertilization on tan spot infections in wheat.
The manuscript suffers from some weaknesses, which that I commented on in the pdf file.
I have several comments that could probably be resolved by minor revision.
The paragraph on resistance is one of the essential parts of the manuscript, but it is not sufficiently discussed.
It would be worth explaining this further.

Author Response
Dear Reviewer 1,
We would like to thank you for spending time reviewing our manuscript. Your time and efforts are highly appreciated. We found that your comments are valuable and improve our manuscript. Please find below a point-by-point response to your comments. Revisions made in the manuscript can be seen as "track changes"
Reviewer comments:
The manuscript presents some exciting novel results that should be published after some editing. This paper addresses research on the combined effects of fungicides and nitrogen fertilization on tan spot infections in wheat.
The manuscript suffers from some weaknesses, which that I commented on in the pdf file.
I have several comments that could probably be resolved by minor revision.
ANSWER. We have revised the entire manuscript considering your comments. Please see the manuscript with revisions attach
The paragraph on resistance is one of the essential parts of the manuscript, but it is not sufficiently discussed. It would be worth explaining this further.
ANSWER. We have revised and expanded this paragraph as suggested (lines 116-120)
Reviewer 2 Report
The manuscript describes interesting research,
The main aspects of this research, which concern the interaction between fungicides/nitrogen and the technological qualities of the grain, have highlighted an important and significant correlation;
The research is relevant, interested in sustainable agriculture;
The research tackled an original topic;
The manuscript is well written and easy to read;
The conclusions are consistent with the evidence and arguments presented.
Author Response
Dear Reviewer,
We would like to thank you for spending time reviewing our manuscript. Your time and efforts are highly appreciated. Revisions made in the manuscript can be seen as "track changes"
Reviewer 3 Report
The manuscript “Fungicide and nitrogen interaction for aboveground biomass from flag leaf emergence and grain yield generation under tan spot infections in wheat” describes an interesting experiment. The objectives of this study were to determine the effects of triple fungicide mixtures containing triazole, strobilurin and carboxamides (TSC) compared to a double triazole-strobilurin mixture (TS) under different N fertilization schemes on the disease severity of tan spot epidemics. While the subject is both interesting and worth publishing in “Plants”,
In my opinion, this is a very interesting study, the manuscript is well-structured, but the writing needs to be improved. Here are some issues should to be addressed to be acceptable, please revised the manuscript carefully:
Comments
· The key words should be distinct from the title words. Authors can change them to be more specific. Please consider updating the keyword list and using synonyms.
· The sentences “ Quantifying the consequences of tan spot epidemics and its interaction with fungicides mixtures and nitrogen (N) rates would be of” please revise the grammar.
· The novelty in study is not clear, please try to improve it in the abstract.
· In the first paragraph please try to enrich it with recent publication in wheat which, discussion the many factors that affect wheat productivity at different aspects, which will really improve the story of the introduction please refer to the following references:
- Temporal complementarity between roots and mycorrhizal fungi drives wheat nitrogen use efficiency. https://doi.org/10.1111/nph.18419
- TaHsfA2-1, a new gene for thermotolerance in wheat seedlings: Characterization and functional roles. 10.1016/j.jplph.2020.153135
- Differentiate effects of non-hydraulic and hydraulic root signaling on yield and water use efficiency in diploid and tetraploid wheat under drought stress. Environmental and Experimental Botany 181, 104287. https://www.sciencedirect.com/science/article/abs/pii/S0098847220303130
- Association of jasmonic acid priming with multiple defense mechanisms in wheat plants under high salt stress. Frontiers in Plant Science 13, 886862.
· “Although the pathogen can harm the crop at any stage of” please use affect instead of harm.
· Please try to improve the hypothesis of the study using the different approaches used in the current study to build a solid hypothesis of this study to be more attractive and clear for the readers.
· Which design used for this experiment, how many factors and their levels used in this study, please provide these information’s under a heading namely Statistical analysis and Experimental design.
· Please try to add one sentence at the end of each paragraph in the results section to conclude the whole paragraph to make it easy to follow.

Author Response
Dear Reviewer,
We would like to thank you for spending time reviewing our manuscript. Your time and efforts are highly appreciated. We found that your comments are valuable and improve our manuscript. Please find below a point-by-point response to your comments. Revisions made in the manuscript can be seen as "track changes"
Reviewer comments:
The manuscript “Fungicide and nitrogen interaction for aboveground biomass from flag leaf emergence and grain yield generation under tan spot infections in wheat” describes an interesting experiment. The objectives of this study were to determine the effects of triple fungicide mixtures containing triazole, strobilurin and carboxamides (TSC) compared to a double triazole-strobilurin mixture (TS) under different N fertilization schemes on the disease severity of tan spot epidemics. While the subject is both interesting and worth publishing in “Plants”,
In my opinion, this is a very interesting study, the manuscript is well-structured, but the writing needs to be improved. Here are some issues should to be addressed to be acceptable, please revised the manuscript carefully:
Comments
- The key words should be distinct from the title words. Authors can change them to be more specific. Please consider updating the keyword list and using synonyms.
ANSWER. We have updated the keywords considering your comments (lines 43-44)
- The sentences “ Quantifying the consequences of tan spot epidemics and its interaction with fungicides mixtures and nitrogen (N) rates would be of” please revise the grammar.
ANSWER. We have revised this sentence as suggested (lines 17-18)
- The novelty in study is not clear, please try to improve it in the abstract.
ANSWER. We have revised the abstract and highlighted the novelty of our study as suggested (lines 20 to 23)
In the first paragraph please try to enrich it with recent publication in wheat which, discussion the many factors that affect wheat productivity at different aspects, which will really improve the story of the introduction please refer to the following references:
- Temporal complementarity between roots and mycorrhizal fungi drives wheat nitrogen use efficiency. https://doi.org/10.1111/nph.18419
- TaHsfA2-1, a new gene for thermotolerance in wheat seedlings: Characterization and functional roles. 10.1016/j.jplph.2020.153135
- Increasing atmospheric CO2 differentially supports arsenite stress mitigating impact of arbuscular mycorrhizal fungi in wheat and soybean plants. Chemosphere 296, 134044. https://doi.org/10.1016/j.chemosphere.2022.134044
- Differentiate effects of non-hydraulic and hydraulic root signaling on yield and water use efficiency in diploid and tetraploid wheat under drought stress. Environmental and Experimental Botany 181, 104287. https://www.sciencedirect.com/science/article/abs/pii/S0098847220303130
- Differentiate responses of tetraploid and hexaploid wheat (Triticum aestivum L.) to moderate and severe drought stress: A cue of wheat domestication. Plant Signaling and Behavior 1839710. https://www.ncbi.nlm.nih.gov/pmc/articles/PMC7781840/
- Association of jasmonic acid priming with multiple defense mechanisms in wheat plants under high salt stress. Frontiers in Plant Science 13, 886862.
ANSWER. We have included a section in the introduction taking into consideration the recommended references (lines 61 to 78)
- “Although the pathogen can harm the crop at any stage of” please use affect instead of harm.
Answer. We replaced this word as suggested (line 86)
- Please try to improve the hypothesis of the study using the different approaches used in the current study to build a solid hypothesis of this study to be more attractive and clear for the readers.
Answer. We have improved the hypothesis of our study (lines 146 to 150)
- Which design used for this experiment, how many factors and their levels used in this study, please provide these information’s under a heading namely Statistical analysis and Experimental design.
Answer. We included information of the experimental design under a new section as suggested (lines 227 to 231)
- Please try to add one sentence at the end of each paragraph in the results section to conclude the whole paragraph to make it easy to follow.
Answer. This detail was corrected across the result section
Round 2
Reviewer 3 Report
Thanks for your contribution,
The authors have successed to address my comments